# Stress on caregivers providing prolonged mechanical ventilation patient care in different facilities: A cross-sectional study

Yeong-Ruey Chu[1], Chin-Jung Liu[2,3], Chia-Chen Chu[2], Pei-Tseng Kung[4‡], Wen-Yu Chou[5], Wen-Chen Tsai[5‡]*

**1** Department of Public Health, China Medical University, Taichung, Taiwan, **2** Department of Respiratory Therapy, China Medical University Hospital, Taichung, Taiwan, **3** School of Nursing, China Medical University, Taichung, Taiwan, **4** Department of Healthcare Administration, Asia University, Taichung, Taiwan, **5** Department of Health Services Administration, China Medical University, Taichung, Taiwan

☯ These authors contributed equally to this work.
‡ P-TK and W-CT also contributed equally to this work.
* wtsai@mail.cmu.edu.tw

**Data Availability Statement:** We have provided the minimal raw data set in the supporting information file (S4 FileS4 File) for the Figs 1, 2, and 3. This study was approved by the Institutional

## Abstract

### Purpose

Taiwan has implemented an integrated prospective payment program (IPP) for prolonged mechanical ventilation (PMV) patients that consists of four stages of care: intensive care unit (ICU), respiratory care center (RCC), respiratory care ward (RCW), and respiratory home care (RHC). We aimed to investigate the life impact on family caregivers of PMV patients opting for a payment program and compared different care units.

### Method

A total of 610 questionnaires were recalled. Statistical analyses were conducted by using the chi-square test and multivariate logistic regression model.

### Results

The results indicated no associations between caregivers' stress levels and opting for a payment program. Participants in the non-IPP group spent less time with friends and family owing to caregiver responsibilities. The results of the family domain show that the RHC group (OR = 2.54) had worsened family relationships compared with the ICU group; however, there was less psychological stress in the RCC (OR = 0.54) and RCW (OR = 0.16) groups than in the ICU group. In the social domain, RHC interviewees experienced reduced friend and family interactivity (OR = 2.18) and community or religious activities (OR = 2.06) than the ICU group. The RCW group felt that leisure and work time had less effect (OR = 0.37 and 0.41) than the ICU group. Furthermore, RCW interviewees (OR = 0.43) were less influenced by the reduced family income than the ICU group in the economic domain.

Review Board of China Medical University Hospital
(IRB No. CMUH102-REC3-105), Taiwan.

**Funding:** This study was supported by grants
(grant numbers DOH102-NH-9009; DMR-109-
015) from the National Health Insurance
Administration and China Medical University
Hospital, Taiwan. The funders had no role in study
design, data collection and analysis, decision to
publish, or preparation of the manuscript.

**Competing interests:** The authors have declared
that no competing interests exist.

**Abbreviations:** PMV, Prolonged Mechanical
Ventilation; IPP, Integrated Payment program; ICU,
intensive care unit; RCC, respiratory care centers;
RCW, respiratory care ward; RHC, Respiratory
home care.

## Conclusions

RHC family caregivers had the highest level of stress, whereas family caregivers in the
RCW group had the lowest level of stress.

## Introduction

Prolonged mechanical ventilation (PMV) is defined as the operation of a mechanical ventila-
tion support system for more than 6 hours per day and exceeding 21 days [1, 2]. The PMV
incidence rates per 100 ICU admissions in Europe and the US were found to be 5%-11% [3–
6]. Taiwan had a PMV rate per 100 mechanical ventilation patients of 20% [7], while China
had a PMV rate of 36% [8], and some PMV patients were found to occupy intensive care unit
(ICU) beds. To increase the emergency department turnover rate, control medical expenses,
and increase the weaning rate of mechanical ventilation, weaning centers (also called subacute
or post-acute care) have been established to rapidly liberate patients from the need for
mechanical ventilators in the United States [9, 10], Europe [4, 11], and Taiwan [12–14].

The Taiwan National Health Insurance provides a comprehensive payment system for inte-
grated prospective payment plans for PMV patients called integrated prospective payment
program (IPP) who are at least 17 years old. This system has integrated payments with man-
aged care systems since July 2000 [2]. This IPP plan covers four types of care and different pay-
ments, including ICU for critical care, respiratory care centers (RCCs) for step-down subacute
units, respiratory care ward (RCW) for long-term care facilities, and respiratory home care
(RHC) for PMV patients [2]. The regulation of IPP are fee-for-service ICU care (within 21 d),
per-diem RCC (for up to 42 d), per capitation RCW, and per-month home ventilation service.
The ICU, RCC, and RCW belong to the institutions, and 24 hours medical staff are available.
The patients who use ventilators at home have nurses and respiratory therapists visit twice a
month and physician visit once two months.

The intervention of the IPP has increased the turnover rate of ICU beds and reduced the
length of hospital stay. However, PMV patients in RCC, RCW, and RHC have seen a signifi-
cant increase in the length of required care [13–15]. Studies that as general home care patients
are more familiar with the environment at home, they feel more comfortable, amenable, and
experience a better than in-house care group [16]. However, it was also found that most PMV
patients were elderly patients [17, 18] with poor life function independence [19] and had a
greater care burden which directly affected their family caregivers [20–24].

Caregivers' fear of caring for PMV patients can cause stress on the physical health, mind,
and soul of both the patients and caregivers [25]. Moreover, it can impose financial burdens
on families. Studies show low willingness in the main caregivers of PMV patients because of
the high pressure that the job puts on them [16, 20, 26]. However, there is a shortage of studies
exploring whether the burden of main family caregivers with patients at different unit has
been affected by the IPP two decades ago. This study aimed to compare the associations
between caregivers' stress levels and opting for a payment program, and compared with the
impaction of life on caregivers of PMV patients in the different units.

## Materials and methods

### Questionnaire development

We invited 10 PMV subject care experts from each type of institution in northern, central, and
southern Taiwan to constitute a panel of experts. The primary family members of the three

PMV patients and three nursing staff members from the same RCW formed a focus group to identify the structure and content of the questionnaire items. The content of the questionnaire included (1) the characteristics of respondents, (2) the characteristics of PMV patients, (3) the effect of care for PMV patients on their families, society, and family finances (S1 File); and (4) relevant information on mechanical ventilation.

The questionnaires have three domains (family, social, and economic). They measured aspects related to their lives by a Likert scale (from strongly agree to strongly disagree) to compare the impact of the IPP and different units of primary caregivers of PMV patients on the stress of life. Then we invited five experts to assess the validity of the questionnaire content following the completion of its design. The average content validity index (CVI) was 0.96 (0.73 to 1.00). The reliability of the questionnaire was measured using the Kuder-Richardson Formula 20 (KR-20). After the questionnaire was tested, the KR-20 coefficient of this study was found to be 0.78.

## Power of sample size

This was a cross-sectional observational study. According to Mohamad Adam Bujang et al.'s recommendation for observational large-sample studies using logistic regression, the minimum sample size needs to be at least 500 [27]. A sufficient total of 601 valid samples were collected in this study.

## Participants

The participants included in this study were at least 20 years of age and served as the main family caregivers of PMV patients from northern, central, and southern Taiwan. According to our definition of the stages of PMV patients, the acute ICU stage was the time in ICU and/or less than 7 days in RCC after transfer from ICU; the subacute stage was $\geq$30 days of stay in RCC; the chronic stage was $\geq$30 days of RCW stay; and finally, the RHC was $\geq$30 days of using a mechanical ventilator at home. The participants were divided into two groups (IPP and none-IPP group) according to whether they had joined IPP.

## Participant consent

Because patients are mostly unconscious, interviewers are required to assist in collecting patients' clinical information, so the research required medical staff, case managers, or respiratory therapists to recruit those who understood the situation of the subjects to agree to do the interview and explain the parameters to the respondents according to the interview instructions (S2 File). This study was approved by the institutional review board of the China Medical University Hospital (IRB No. CMUH102-REC3-105).

## Statistical analysis

Chi-square tests were used to compare the characteristics of patients with PMV in terms of subject characteristics, mechanical ventilation status, and the difference between the impact on the main caregivers of patients at different stages. Finally, we combined "strongly agree," "agree" and "no objection" into one group and "disagree" and "strongly disagree" into another group, and used the logistic regression model to compare the differences in stress levels experienced by the main family caregivers for ICU, RCC, RCW, and RHC patients. Statistical significance was set at $p < 0.05$, and SAS (version 9.4, SAS Institute Inc., Cary, NC, USA) was used for the analysis.

## Results

### Respondent and patient characteristics

As comprehensive analyses datasets were generated by a previous study [28], including 687 eligible respondents from 64 institutions (6 medical centers, 11 regional hospitals, 23 district hospitals, 15 nursing homes, and 9 home care centers) (S3 File). The study was conducted from November 1, 2013, to April 15, 2014. A total of 601 questionnaires were completed (effective questionnaire response rate 87%) and 50.75% were women, with an average age of 51.88 years. A majority of the responses were provided by children of PMV parents (54.74%). The monthly expense paid by the family for the IPP group was significantly lower than that for the non-IPP group (USD 745 vs. USD 912 respectively) (Table 1). The average age of the PMV patients was 70.76 years. Only 38.77% of patients were in a state of alert consciousness, with 34.78% with near-terminal illnesses. A total of 89.35% of the patients had daily mechanical ventilation of 19–24 hours (Table 2).

### Impact of caring for PMV patients on families

Significantly more caregivers in the non-IPP group ($p$ = 0.039), compared with the IPP group, reported reduced time with friends and family. In addition, there were no significant statistical differences in other factors (Table 3).

### Impact of socioeconomic conditions, caregiver occupation, underlying diseases of the patients on caregiver stress levels and family relationships

We used family income and education level to represent the socioeconomic conditions of caregivers. We then compared the family income and education level of the caregivers of PMV patients at four stages with the psychological pressure and family relationship using Spearman's rank correlation. In addition, we used the Chi-square test to analyze the patient's underlying disease, psychological pressure on the caregiver, and family relationships. The results showed that there was no significant difference in the family income of caregivers between psychological stress and family relationships. However, we found that the higher the education level, the lower the adverse effect on the relationship between family members; moreover, this difference was statistically significant ($p<0.05$) (see Table 4). There was no significant correlation between the underlying disease of PMV patients and the psychological stress on caregivers and family relationships (Table 5).

Regarding the impact on the family caregivers' life among the ICU, RCC, RCW and RHC groups, results showed that respondents from the RHC group showed more agreement to the item, "Taking care of the patient worsens the relationship between family members" than interviewees in the ICU group (OR = 2.54 (95%CI 1.60–4.05), $p<0.001$); however, respondents from the RCW group showed less agreement to the item, "I experienced physical stress because caring for the patient" than those in the ICU group (OR = 0.61 (95%CI 0.38–1.00), $p$ = 0.05). For the item, "I feel psychologically stressed from caring for the patient," respondents from the RCC (OR = 0.54 (95%CI 0.29–0.99), $p$ = 0.048) and RCW (OR = 0.46 (95%CI 0.25–0.85), $p$ = 0.012) groups showed less agreement than those from the ICU group (Fig 1) (S4 File).

In the social domain, for items such as "I feel that my time for my friends and family has reduced because of caring for the patient" and "I feel that the time for community or religious activities has reduced because of caring for the patient," caregivers in the RHC group showed more agreement than those in the ICU group (OR = 2.18 (95%CI 1.21–3.92), $p$ = 0.01 and OR = 2.06 (95%CI 1.15–3.68), $p$ = 0.014, respectively). For items such as "I feel that leisure time has decreased because of caring for the patient," the RCC and RCW groups were in less

**Table 1. Demographic characteristics of respondents with and without IPP.**

| Variables | Total N = 601 | | Non-IPP: N = 207 | | IPP: N = 394 | | p value |
|---|---|---|---|---|---|---|---|
| | N | % | N | % | N | % | |
| Gender | | | | | | | 0.674 |
| Male | 296 | 49.25 | 99 | 47.83 | 197 | 50.00 | |
| Female | 305 | 50.75 | 108 | 52.17 | 197 | 50.00 | |
| Age in years | | | | | | | |
| Mean age (SD) | 51.88 | (11.84) | 53.30 | (11.59) | 51.13 | (11.92) | 0.032# |
| Educational level | | | | | | | 0.504 |
| None | 13 | 2.16 | 7 | 3.38 | 6 | 1.52 | |
| ≦Grade 9 | 161 | 26.79 | 53 | 25.60 | 108 | 27.41 | |
| High school to college | 403 | 67.05 | 139 | 67.15 | 264 | 67.01 | |
| Graduate school | 24 | 3.99 | 8 | 3.86 | 16 | 4.06 | |
| Married | | | | | | | 0.636 |
| Yes | 480 | 80.00 | 170 | 82.13 | 310 | 78.88 | |
| Never | 90 | 15.00 | 28 | 13.53 | 62 | 15.78 | |
| Been married | 30 | 5.00 | 9 | 4.35 | 21 | 5.34 | |
| Monthly salary in USD | | | | | | | 0.194 |
| <1000 | 154 | 25.71 | 45 | 21.84 | 109 | 27.74 | |
| 1000~2000 | 235 | 39.23 | 94 | 45.63 | 141 | 35.88 | |
| 2000~3000 | 125 | 20.87 | 42 | 20.39 | 83 | 21.12 | |
| 3000~4000 | 51 | 8.51 | 15 | 7.28 | 36 | 9.16 | |
| ≥4000 | 34 | 5.68 | 10 | 4.85 | 24 | 6.11 | |
| Religion | | | | | | | 0.746 |
| Yes | 502 | 83.53 | 171 | 82.61 | 331 | 84.01 | |
| No | 99 | 16.47 | 36 | 17.39 | 63 | 15.99 | |
| Relationship with PMV patient | | | | | | | 0.423 |
| Parents | 41 | 6.82 | 8 | 3.86 | 33 | 8.38 | |
| Couple | 111 | 18.47 | 42 | 20.29 | 69 | 17.51 | |
| Children | 329 | 54.74 | 115 | 55.56 | 214 | 54.31 | |
| Children-in-law | 59 | 9.82 | 21 | 10.14 | 38 | 9.64 | |
| Brothers and sisters | 27 | 4.49 | 10 | 4.83 | 17 | 4.31 | |
| Grandchildren | 17 | 2.83 | 7 | 3.38 | 10 | 2.54 | |
| Other | 17 | 2.83 | 4 | 1.93 | 13 | 3.30 | |
| Can someone take turns taking care of the patient with you? | | | | | | | 0.809 |
| No | 194 | 32.28 | 65 | 31.40 | 129 | 32.74 | |
| Yes | 407 | 67.72 | 142 | 68.60 | 265 | 67.26 | |
| Average monthly expense (USD) Mean (SD) | 801.69 | (902.27) | 912.61 | (601.97) | 745.3553 | (1017.66) | 0.014 |

IPP: integrated prospective payment program.

One US dollar (USD) was 30 New Taiwan dollars (NTD) on Sep. 2013.

# t-test.

agreement than those in the ICU group (RCC, OR = 0.47 (95%CI 0.26–0.86), $p$ = 0.014 and RCW, OR = 0.37 (95%CI 0.21–0.67, $p$<0.001). For the item, "My work is affected because of caring for the patient." the RCW group was in less agreement than the ICU group (OR = 0.41 (95%CI 0.24–0.69), $p$<0.001) (Fig 2) (S4 File).

In the economic domain, the ICU group (OR = 0.43 (95%CI 0.26–0.72), $p$ = 0.001) (Fig 3) (S4 File) had a greater level of agreement with the item "Reduced family income due to inability to work owing to caregiving responsibilities," than the RCW group.

**Table 2. Demographic characteristics of PMV patients with and without IPP.**

| Variables | Total N = 601 | | Non-IPP: N = 207 | | IPP: N = 394 | | P value |
|---|---|---|---|---|---|---|---|
| | N | % | N | % | N | % | |
| Gender | | | | | | | 1.000 |
| Male | 304 | 50.58 | 105 | 50.72 | 199 | 50.51 | |
| Female | 297 | 49.42 | 102 | 49.28 | 195 | 49.49 | |
| Age in years | | | | | | | |
| Mean age (SD) | 70.76 | (17.23) | 73.15 | (15.99) | 69.51 | (17.73) | 0.014# |
| Conscious status | | | | | | | 0.263 |
| Coma | 194 | 32.28 | 66 | 31.88 | 128 | 32.49 | |
| Unconsciousness | 174 | 28.95 | 68 | 32.85 | 106 | 26.90 | |
| Alert | 233 | 38.77 | 73 | 35.27 | 160 | 40.61 | |
| Cause of respiratory failure | | | | | | | **0.003** |
| Chronic lung disease | 131 | 21.80 | 51 | 24.64 | 80 | 20.30 | |
| Central neuropathy | 185 | 30.78 | 60 | 28.99 | 125 | 31.73 | |
| Catastrophic illnesses | 209 | 34.78 | 83 | 40.10 | 126 | 31.98 | |
| Other | 76 | 12.65 | 13 | 6.28 | 63 | 15.99 | |
| Unit | | | | | | | **<0.001** |
| ICU | 150 | 24.96 | 60 | 28.99 | 90 | 22.84 | |
| RCC | 150 | 24.96 | 60 | 28.99 | 90 | 22.84 | |
| RCW | 150 | 24.96 | 60 | 28.99 | 90 | 22.84 | |
| RHC | 151 | 25.12 | 27 | 13.04 | 124 | 31.47 | |
| Daily bed-time (hrs.) | | | | | | | **0.004$** |
| 0–6 | 1 | 0.17 | 1 | 0.48 | 0 | 0.00 | |
| 7–12 | 37 | 6.16 | 5 | 2.42 | 32 | 8.12 | |
| 13–18 | 26 | 4.33 | 6 | 2.90 | 20 | 5.08 | |
| 19–24 | 537 | 89.35 | 195 | 94.20 | 342 | 86.80 | |
| Joint IPP and waive copayment (yes) | 494 | 82.20 | 143 | 69.08 | 351 | 89.09 | **<0.001** |

# t-test

$ Fisher's exact test; PMV: prolonged mechanical ventilation; IPP: integrated prospective payment program; ICU: intensive care unit; RCC: respiratory care center; RCW: respiratory care ward; RHC; respiratory home care.

## Discussion

The PMV patients and respondents had an average age of 70.76 and 51.88 years respectively. Of the main caregivers, 54.74% were children, and 80% of them were married. In the IPP group, the monthly expenses spent on patients were significantly lower than those of the non-IPP group. The caregivers of home-based patients experienced higher stress levels than caregivers of patients in another stage. The caregivers of ICU patients also experienced high stress levels, but only second to those taking care of home-based patients. Furthermore, it was found that caregivers of RCW patients had significantly lower stress levels than the family caregivers of patients in other situations.

 This study demonstrated that the stress levels experienced by caregivers of home-based PMV patients (RHC) were 1.21–2.54 times that of their counterparts who care for ICU patients. According to the results, caregivers of home-based PMV patients had poorer sleep quality [29], higher risks of depression, and poorer health conditions [30, 31]. We also found that although caregivers of home-based patients had higher stress levels than those of RCW patients [29], the difference was not significant. As demonstrated in our study, the stress level of the caregivers of home-based patients was the highest, followed by that of caregivers of ICU,

**Table 3. Comparison of the impact of IPP for PMV patients on the life of caregivers.**

| Variables | All | | Non-IPP | | IPP | | p value |
|---|---|---|---|---|---|---|---|
| | N | % | N | % | N | % | |
| **Total** | 601 | 100 | 207 | 34.44 | 394 | 65.56 | |
| **[Family domain]** | | | | | | | |
| **Taking care of the patient worsens the relationship between family members.** | | | | | | | **0.199** |
| strongly disagree | 112 | 18.64 | 37 | 17.87 | 75 | 19.04 | |
| disagree | 227 | 37.77 | 81 | 39.13 | 146 | 37.06 | |
| no objection | 91 | 15.14 | 37 | 17.87 | 54 | 13.71 | |
| agree | 133 | 22.13 | 36 | 17.39 | 97 | 24.62 | |
| strongly agree | 38 | 6.32 | 16 | 7.73 | 22 | 5.58 | |
| **I feel that family life is affected because of caring for the patient** | | | | | | | **0.195** |
| strongly disagree | 47 | 7.82 | 14 | 6.76 | 33 | 8.38 | |
| disagree | 121 | 20.13 | 53 | 25.60 | 68 | 17.26 | |
| no objection | 76 | 12.65 | 25 | 12.08 | 51 | 12.94 | |
| agree | 285 | 47.42 | 91 | 43.96 | 194 | 49.24 | |
| strongly agree | 72 | 11.98 | 24 | 11.59 | 48 | 12.18 | |
| **I experience physical stress because of caring for the patient** | | | | | | | **0.051** |
| strongly disagree | 46 | 7.65 | 12 | 5.80 | 34 | 8.63 | |
| disagree | 132 | 21.96 | 59 | 28.50 | 73 | 18.53 | |
| no objection | 98 | 16.31 | 28 | 13.53 | 70 | 17.77 | |
| agree | 242 | 40.27 | 79 | 38.16 | 163 | 41.37 | |
| strongly agree | 83 | 13.81 | 29 | 14.01 | 54 | 13.71 | |
| **I feel psychologically stressed from caring for the patient** | | | | | | | **0.651** |
| strongly disagree | 27 | 4.49 | 9 | 4.35 | 18 | 4.57 | |
| disagree | 83 | 13.81 | 32 | 15.46 | 51 | 12.94 | |
| no objection | 56 | 9.32 | 22 | 10.63 | 34 | 8.63 | |
| agree | 306 | 50.92 | 97 | 46.86 | 209 | 53.05 | |
| strongly agree | 129 | 21.46 | 47 | 22.71 | 82 | 20.81 | |
| **[Social domain]** | | | | | | | |
| **I feel that my time for my friends and family has reduced because of caring for the patient** | | | | | | | **0.039** |
| strongly disagree | 36 | 5.99 | 12 | 5.80 | 24 | 6.09 | |
| disagree | 126 | 20.97 | 56 | 27.05 | 70 | 17.77 | |
| no objection | 121 | 20.13 | 31 | 14.98 | 90 | 22.84 | |
| agree | 247 | 41.10 | 82 | 39.61 | 165 | 41.88 | |
| strongly agree | 71 | 11.81 | 26 | 12.56 | 45 | 11.42 | |
| **I feel that the time for community or religious activities has reduced because of caring for the patient** | | | | | | | **0.170** |
| strongly disagree | 37 | 6.16 | 13 | 6.28 | 24 | 6.09 | |
| disagree | 124 | 20.63 | 53 | 25.60 | 71 | 18.02 | |
| no objection | 144 | 23.96 | 50 | 24.15 | 94 | 23.86 | |
| agree | 225 | 37.44 | 66 | 31.88 | 159 | 40.36 | |
| strongly agree | 71 | 11.81 | 25 | 12.08 | 46 | 11.68 | |
| **I feel that leisure time has decreased because of caring for the patient** | | | | | | | **0.125** |
| strongly disagree | 28 | 4.66 | 9 | 4.35 | 19 | 4.82 | |
| disagree | 88 | 14.64 | 39 | 18.84 | 49 | 12.44 | |
| no objection | 95 | 15.81 | 38 | 18.36 | 57 | 14.47 | |
| agree | 286 | 47.59 | 88 | 42.51 | 198 | 50.25 | |
| strongly agree | 104 | 17.30 | 33 | 15.94 | 71 | 18.02 | |
| **My work is affected because of caring for the patient** | | | | | | | **0.392** |

*(Continued)*

**Table 3.** (Continued)

| Variables | All | | Non-IPP | | IPP | | p value |
|---|---|---|---|---|---|---|---|
| | N | % | N | % | N | % | |
| strongly disagree | 34 | 5.66 | 13 | 6.28 | 21 | 5.33 | |
| disagree | 113 | 18.8 | 47 | 22.71 | 66 | 16.75 | |
| no objection | 118 | 19.63 | 41 | 19.81 | 77 | 19.54 | |
| agree | 213 | 35.44 | 66 | 31.88 | 147 | 37.31 | |
| strongly agree | 123 | 20.47 | 40 | 19.32 | 83 | 21.07 | |
| **It is difficult to find proper social support or assistance to take care of the patient** | | | | | | | **0.085** |
| strongly disagree | 37 | 6.16 | 9 | 4.35 | 28 | 7.11 | |
| disagree | 149 | 24.79 | 63 | 30.43 | 86 | 21.83 | |
| no objection | 132 | 21.96 | 40 | 19.32 | 92 | 23.35 | |
| agree | 206 | 34.28 | 65 | 31.40 | 141 | 35.79 | |
| strongly agree | 77 | 12.81 | 30 | 14.49 | 47 | 11.93 | |
| **[Economic domain]** | | | | | | | |
| **Reduced family income due to inability to work owing to caregiving responsibilities** | | | | | | | **0.344** |
| strongly disagree | 42 | 6.99 | 11 | 5.31 | 31 | 7.87 | |
| disagree | 110 | 18.30 | 46 | 22.22 | 64 | 16.24 | |
| no objection | 113 | 18.80 | 40 | 19.32 | 73 | 18.53 | |
| agree | 216 | 35.94 | 72 | 34.78 | 144 | 36.55 | |
| strongly agree | 120 | 19.97 | 38 | 18.36 | 82 | 20.81 | |
| **I am under financial pressure because of the cost of caring for the patient** | | | | | | | **0.446** |
| strongly disagree | 21 | 3.49 | 7 | 3.38 | 14 | 3.55 | |
| disagree | 62 | 10.32 | 28 | 13.53 | 34 | 8.63 | |
| no objection | 105 | 17.47 | 33 | 15.94 | 72 | 18.27 | |
| agree | 260 | 43.26 | 88 | 42.51 | 172 | 43.65 | |
| strongly agree | 153 | 25.46 | 51 | 24.64 | 102 | 25.89 | |

IPP: integrated prospective payment program; PMV: prolonged mechanical ventilation.

RCC, and RCW patients. Therefore, apart from continued attention paid to the stress of the main caregivers of home-based PMV patients [20, 22, 29, 30, 32, 33], we cannot ignore the stress experienced by caregivers of ICU patients.

The results showed that caregivers of ICU and RHC patients had higher stress levels than those of RCC and RCW patients. This might be caused by uncertainties in emergency medical treatments performed to save the lives of ICU patients [34–37] and the pressure to make life-making decisions on behalf of the patients [38]. These caregivers may have a higher risk of developing depression [39] because their stress levels increase with the severity of the patients' condition [40]. Although RCC and RCW patients were mechanical ventilator users, they had already experienced the uncertainty of the acute phase. At this point, their family members

**Table 4. The socioeconomic conditions and occupation of caregiver's impact on stress level and family relationship.**

| Variables | Family income of caregiver | | Education of caregiver | |
|---|---|---|---|---|
| | correlation coefficient | p-value[1] | correlation coefficient | p-value[1] |
| Family relationship | -0.017 | 0.677 | -0.166 | <0.001 |
| Psychologically stressed | 0.008 | 0.850 | -0.080 | 0.050 |

[1]Spearman's rank correlation test.

**Table 5. Association of underlying diseases of the patients with stress level and family relationship.**

| Underline disease | Chronic pulmonary disease | | Central neuropathy | | Catastrophic illness | | Others | | P-value[1] |
|---|---|---|---|---|---|---|---|---|---|
| | n | % | n | % | N | % | n | % | |
| **Psychologically stressed** | | | | | | | | | **0.515** |
| strongly disagree | 4 | 3.05 | 9 | 4.86 | 9 | 4.31 | 5 | 6.58 | |
| disagree | 17 | 12.98 | 22 | 11.89 | 37 | 17.70 | 7 | 9.21 | |
| no objection | 12 | 9.16 | 12 | 6.49 | 22 | 10.53 | 10 | 13.16 | |
| agree | 67 | 51.15 | 99 | 53.51 | 104 | 49.76 | 36 | 47.37 | |
| strongly agree | 31 | 23.66 | 43 | 23.24 | 37 | 17.70 | 18 | 23.68 | |
| Total | 131 | 21.80 | 185 | 30.78 | 209 | 34.78 | 76 | 12.65 | |
| **Family relationship** | | | | | | | | | **0.129** |
| strongly disagree | 19 | 14.50 | 38 | 20.54 | 38 | 18.18 | 17 | 22.37 | |
| disagree | 51 | 38.93 | 64 | 34.59 | 89 | 42.58 | 23 | 30.26 | |
| no objection | 19 | 14.50 | 22 | 11.89 | 39 | 18.66 | 11 | 14.47 | |
| agree | 32 | 24.43 | 50 | 27.03 | 34 | 16.27 | 17 | 22.37 | |
| strongly agree | 10 | 7.63 | 11 | 5.95 | 9 | 4.31 | 8 | 10.53 | |
| Total | 131 | 21.80 | 185 | 30.78 | 209 | 34.78 | 76 | 12.65 | |

[1]**Chi-square test.**

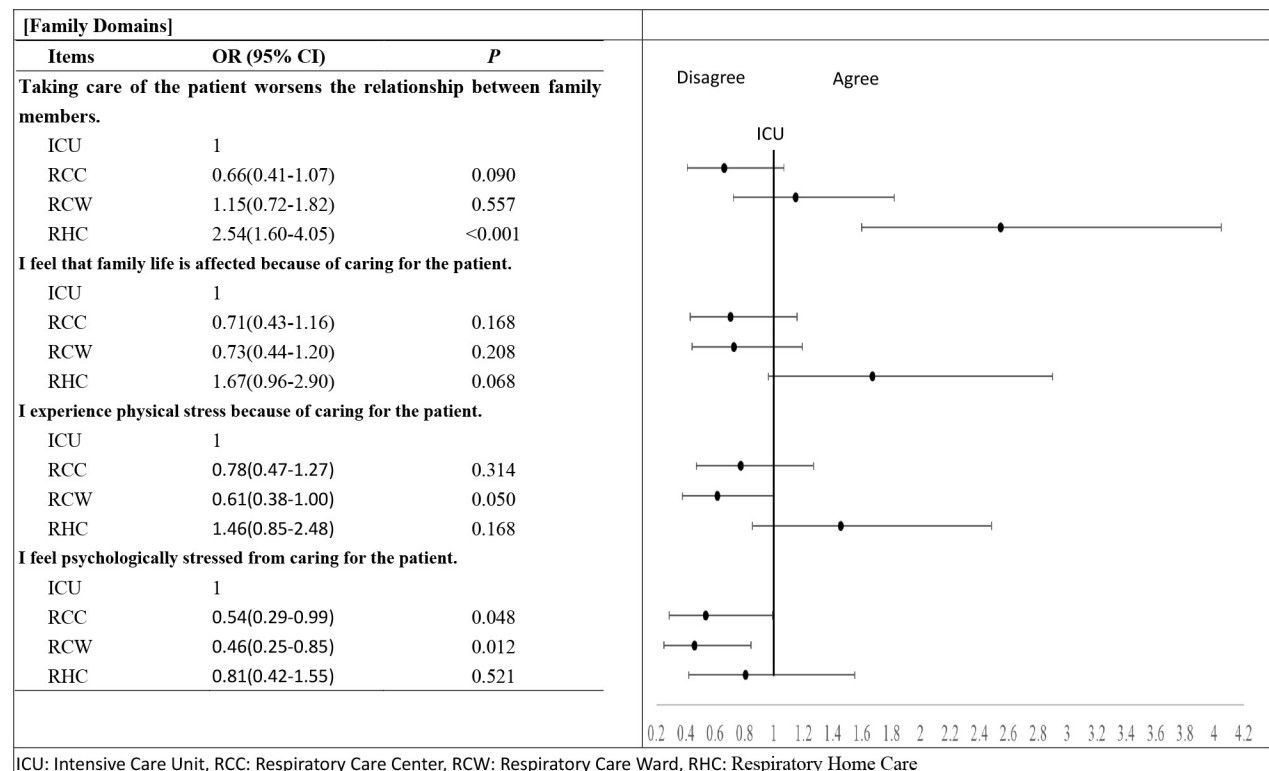

**Fig 1. Odds of agreement in the life impact of the caregivers of PMV patients at various stages at family domain.**

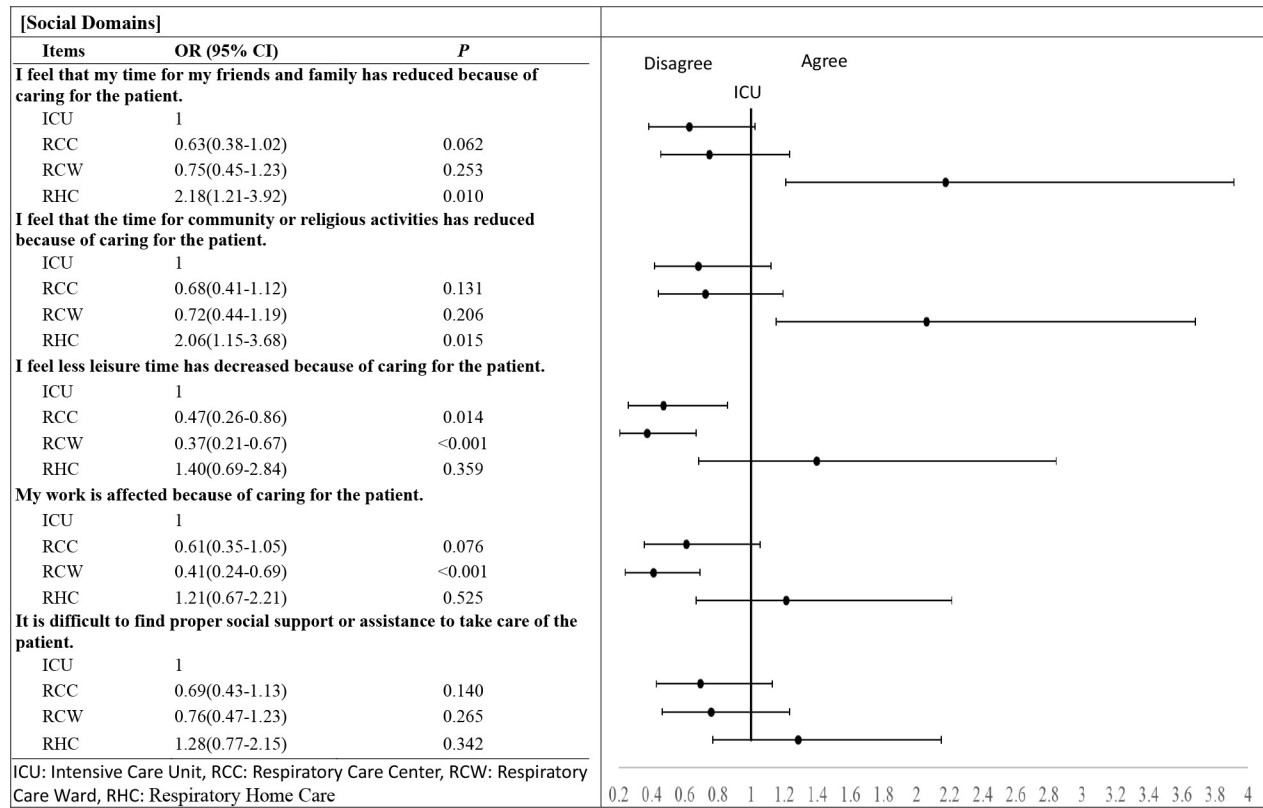

**Fig 2. Odds of agreement in the life impact of the caregivers of PMV patients at various stages at social domain.**

could support each other in caring for patients using only shared knowledge for methods of care. As the caregivers no longer needed to attend to patients 24 hours a day, the stress they experienced was significantly lower than that experienced by caregivers of home-based patients [29].

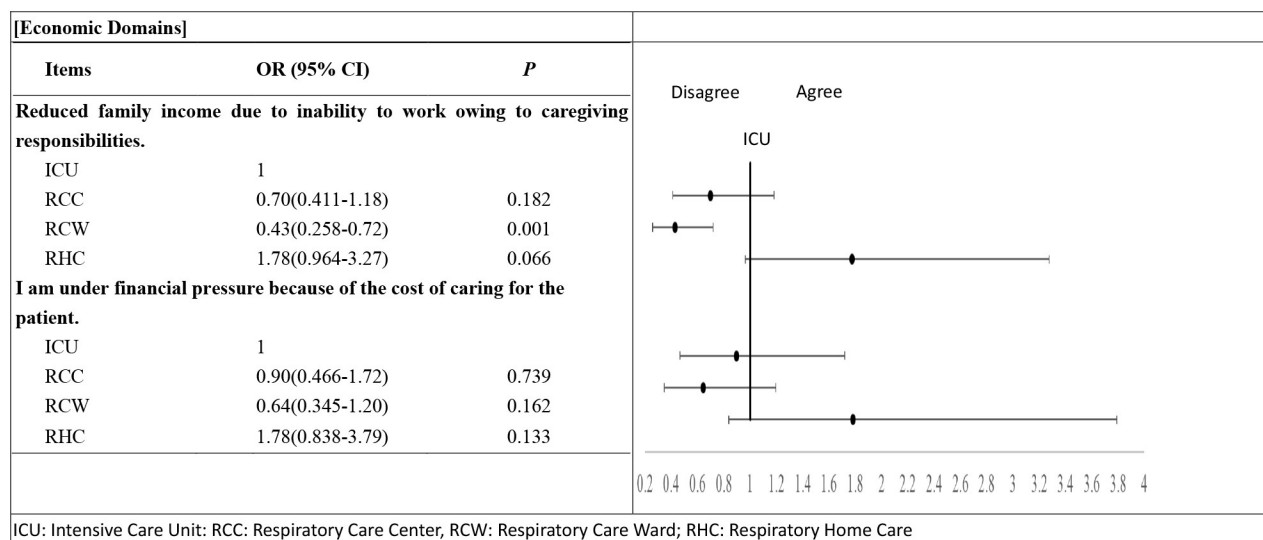

**Fig 3. Odds of agreement in the life impact of the caregivers of PMV patients at various stages at economic domain.**

Since the main caregivers of home-based PMV patients needed to take care of the patients, they had to give up part of their time with family, their comforts of life, and their opportunities to communicate and build relationships with friends and relatives [41]. Notwithstanding, patients with PMV have a poor prognosis [42]. The considerable expense and physical fatigue can aggravate psychological and physical stress experienced by family members, as well as increase socioeconomic burdens over time [21–23, 31, 34]. As a result, most caregivers of PMV patients choose to work with medical institutions instead of taking total care of patients themselves [29, 43]. The caregivers of RCW patients had significantly lower stress levels than those of patients in other situations, indicating that medical institutions were of great help in relieving caregivers' stress. Therefore, PMV care centers in medical institutions can help family caregivers get plenty of rest and reduce the stress they experience [34].

The strengths of our study conducted a large number of participants from 64 institutions. However, some limitations were considered. First, the impact of the socioeconomic condition of participants on stress level on family is not clear. Second, we used income and education to represent the socioeconomic condition which might not be a complete evaluation.

## Conclusions

Family caregivers of RHC patients had the highest level of stress, followed by their counterparts who care for ICU patients; further, the stress levels of caregivers are not associated with IPP. Higher the education level, the lower the adverse effect on the relationship between family members. Future studies to investigate the relationship between socioeconomic condition and stress level is suggested.

## Supporting information

**S1 File. Question about the impact of taking care of prolonged mechanical ventilation patients on life (English and Chinese version).**
(DOCX)

**S2 File. Interviewees manual (English and Chinese version).**
(DOCX)

**S3 File. 64 Interviewee's institutions information.**
(DOCX)

**S4 File. The impact on the family caregivers' life between the ICU, RCC, RCW and RHC groups.**
(DOCX)

## Acknowledgments

We are grateful to these 64 participants of hospitals/institutions for providing administrative support. We are also grateful to Health Data Science Center, China Medical University Hospital for providing administrative and technical support.

## Author Contributions

**Conceptualization:** Pei-Tseng Kung, Wen-Chen Tsai.

**Data curation:** Chin-Jung Liu, Chia-Chen Chu, Pei-Tseng Kung, Wen-Chen Tsai.

**Formal analysis:** Yeong-Ruey Chu, Chin-Jung Liu, Chia-Chen Chu, Pei-Tseng Kung, Wen-Yu Chou, Wen-Chen Tsai.

**Funding acquisition:** Pei-Tseng Kung, Wen-Chen Tsai.

**Investigation:** Yeong-Ruey Chu, Chin-Jung Liu, Chia-Chen Chu, Pei-Tseng Kung, Wen-Yu Chou, Wen-Chen Tsai.

**Methodology:** Yeong-Ruey Chu, Chia-Chen Chu, Pei-Tseng Kung, Wen-Yu Chou, Wen-Chen Tsai.

**Project administration:** Yeong-Ruey Chu, Chia-Chen Chu, Pei-Tseng Kung, Wen-Chen Tsai.

**Resources:** Yeong-Ruey Chu, Chia-Chen Chu, Pei-Tseng Kung, Wen-Yu Chou, Wen-Chen Tsai.

**Software:** Pei-Tseng Kung, Wen-Chen Tsai.

**Supervision:** Pei-Tseng Kung, Wen-Chen Tsai.

**Validation:** Yeong-Ruey Chu, Chin-Jung Liu, Pei-Tseng Kung, Wen-Yu Chou, Wen-Chen Tsai.

**Visualization:** Pei-Tseng Kung, Wen-Chen Tsai.

**Writing – original draft:** Yeong-Ruey Chu, Chin-Jung Liu, Pei-Tseng Kung, Wen-Chen Tsai.

**Writing – review & editing:** Yeong-Ruey Chu, Chin-Jung Liu, Pei-Tseng Kung, Wen-Chen Tsai.

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
