## [Decision Letter · Decision Letter 0]

9 Aug 2021

PONE-D-21-24193

Stress of caregivers providing prolonged mechanical ventilation patient care in different facilities: A cross-sectional study

PLOS ONE

Dear Dr. Tsai,

Thank you for submitting your manuscript to PLOS ONE. After careful consideration, we feel that it has merit but does not fully meet PLOS ONE’s publication criteria as it currently stands. Therefore, we invite you to submit a revised version of the manuscript that addresses the points raised during the review process.

Please address the issues and revise accordingly.

We look forward to receiving your revised manuscript.

Kind regards,

Academic Editor

PLOS ONE

Journal Requirements:

Furthermore, please provide additional information regarding how participants were recruited for the study, and please ensure that you have provided sufficient detail to allow your work to be replicated. 

Moreover,  in your Methods section, please provide a justification for the sample size used in your study, including any relevant power calculations (if applicable).

Finally, please include in your Methods section (or in Supplementary Information files) the participating hospitals/institutions.

3. Please provide additional details regarding participant consent. In the ethics statement in the Methods and online submission information, please ensure that you have specified whether consent was written or verbal/oral. If consent was verbal/oral, please specify: 1) whether the ethics committee approved the verbal/oral consent procedure, 2) why written consent could not be obtained, and 3) how verbal/oral consent was recorded. If your study included minors, please state whether you obtained consent from parents or guardians in these cases. If the need for consent was waived by the ethics committee, please include this information.

5. Please include your tables as part of your main manuscript and remove the individual files. Please note that supplementary tables (should remain/ be uploaded) as separate "supporting information" files

Reviewers' comments:

Reviewer's Responses to Questions

**Comments to the Author**

1. Is the manuscript technically sound, and do the data support the conclusions?

Reviewer #1: Yes

Reviewer #2: Partly

2. Has the statistical analysis been performed appropriately and rigorously? 

Reviewer #1: Yes

Reviewer #2: I Don't Know

3. Have the authors made all data underlying the findings in their manuscript fully available?

Reviewer #1: Yes

Reviewer #2: No

4. Is the manuscript presented in an intelligible fashion and written in standard English?

Reviewer #1: Yes

Reviewer #2: No

5. Review Comments to the Author

Reviewer #1: The article is interesting one because it has looked issues from different aspects rather than medical only. It would be better if you could find out socioeconomic conditions and occupation of caregivers and underlying diseases of the patients impact on stress level and family relationship.

Page 11, last paragraph, I think it's home-based PMV rather than PMV alone.

Overall, the article is good one. Well done

Reviewer #2: missing tables

can not make the review

Many language errors

Many abbreviation don’t correspond correctly to their phrases

Like “RCW” and “RHC”

“The main purpose of this study is to provide a solution for the

shortage of ICU beds”

That is a confusing sentence as this isn’t the aim of the study!!

6. PLOS authors have the option to publish the peer review history of their article (what does this mean?). If published, this will include your full peer review and any attached files.

Reviewer #1: **Yes: **Prof. Shital Adhikari

Reviewer #2: **Yes: **aljamaan, fadi

---

## [Author Response · Author response to Decision Letter 0]

7 Oct 2021

Response to Reviewers’ Comments

PLOS ONE

Ref. No. PONE-D-21-24193

Title: Stress on caregivers providing prolonged mechanical ventilation patient care in different facilities: A cross-sectional study

Dear Editor and reviewers:

We appreciated your helpful comments to improve the presentation of the paper. We have responded to all advice point by point and revised the manuscript. It is our sincere hope that this revision will enhance readability with strengthened discussions to satisfy the requirements of this prestigious journal. 

1. Please include additional information regarding the survey or questionnaire used in the study and ensure that you have provided sufficient details that others could replicate the analyses. For instance, if you developed a questionnaire as part of this study and it is not under a copyright more restrictive than CC-BY, please include a copy, in both the original language and English, as Supporting Information.

Response: Thanks for your comments. 

We provided the questionnaire in both Chinese and English (Attachment 1) and all of the questions are listed in table 3 on the main document (see page 9-11 on table 3).

2. Furthermore, please provide additional information regarding how participants were recruited for the study, and please ensure that you have provided sufficient detail to allow your work to be replicated. 

Response: Thanks for your comments. 

The researcher invited medical staff, case managers, or respiratory therapists to recruit those who fit the criteria of the subjects to accept the interview and explained to the respondents according to the interview instructions (Attachment 2). If the interviewee did not agree to accept the questionnaire survey, they could withdraw their consent from this research without any reason, with no negative consequences.

3. Moreover, in the Methods section, please provide a justification for the sample size used in your study, including any relevant power calculations (if applicable).

Response: Thanks for your queries. 

Our justification for the sample size is as follows, and was edited in methods section.

This was a cross-sectional observational study. According to Mohamad Adam Bujang et al.'s recommendation for observational large-sample studies using logistic regression, the minimum sample size needs to be at least 500 [25]. A total of 601 valid samples were collected in this study, so there were sufficient samples (See line 103-107 on page 5).

We cited the reference 25 as follows: 

Bujang MA, Sa'at N, Sidik T, Joo LC. Sample Size Guidelines for Logistic Regression from Observational Studies with Large Population: Emphasis on the Accuracy Between Statistics and Parameters Based on Real Life Clinical Data. The Malaysian journal of medical sciences: MJMS. 2018;25(4):122-30. Epub 2019/03/28. doi: 10.21315/mjms2018.25.4.12. PubMed PMID: 30914854; PubMed Central PMCID: PMCPMC6422534.

4. Finally, please include in the Methods section (or in Supplementary Information files) of the participating hospitals/institutions.

Response: Thanks for your suggestion. 

We have uploaded the list of participating hospitals/institutions in supplementary information file (the Attachment 3) and provides a list of 64 hospital/institution names.

5. Please provide additional details regarding participant consent. In the ethics statement in the Methods and online submission information, please ensure that you have specified whether consent was written or verbal/oral. If consent was verbal/oral, please specify: 1) whether the ethics committee approved the verbal/oral consent procedure, 2) why written consent could not be obtained, and 3) how verbal/oral consent was recorded. If your study included minors, please state whether you obtained consent from parents or guardians in these cases. If the need for consent was waived by the ethics committee, please include this information.

Response: Thanks for your queries.

Because patients are mostly unconscious and interviewers are required to assist in collecting patients’ clinical information, the research invites medical staff, case managers, or respiratory therapists to recruit those who understand the situation of the patients to do the interview, and explains to the main caregiver of the family according to the interview instructions (see the Attachment 2).

6. We note that the grant information you provided in the ‘Funding Information’ and ‘Financial Disclosure’ sections do not match. When you resubmit, please ensure that you provide the correct grant numbers for the awards you received for your study in the ‘Funding Information’ section.

Response: Thanks for your suggestion. 

This study was supported by grants (grant numbers DOH102-NH-9009; DMR-109-015) from the National Health Insurance Administration and China Medical University Hospital, Taiwan. The funders had no role in study design, data collection and analysis, decision to publish, or preparation of the manuscript (See line 284-288 on page 19).

7. Please include your tables as part of your main manuscript and remove the individual files. Please note that supplementary tables (should remain/ be uploaded) as separate "supporting information" files

Response: Thank you for your comments.

We have inserted tables in the manuscript.

Reviewer #1 

1. The article is interesting one because it has looked issues from different aspects rather than medical only. It would be better if you could find out socioeconomic conditions and occupation of caregivers and underlying diseases of the patients impact on stress level and family relationship.

Response: We used family income and education level to represent the socioeconomic conditions of caregivers, and compared the family income and education level of the caregivers of PMV patients at four stages with the psychological pressure and family relationship with Spearman's rank correlation. In addition, we used the Chi-square test to analyze the patient's underlying disease, psychological pressure of the caregiver, and family relationships. The results showed that there was no significant difference in the family income of caregivers between psychological stress and family relationships, but it was found that the higher the education level, the lower the adverse effect on the relationship between family members, and there was a statistically significant difference (p<0.05) (Table 4). There was no significant correlation between the underlying disease of PMV patients and the psychological stress of caregivers and family relationships (Tables 5). We have revised the text in the manuscript (See table 4-5 on page 14-15.)

2. Page 11, last paragraph, I think it's home-based PMV rather than PMV alone.

Response: Thank you for your comments. 

I really want to express home-base, and modified it as follows:

“The results show that caregivers of ICU and home-based PMV patients have higher stress levels than those of RCC and RCW patients.”

Reviewer #2: 

1. missing tables can not make the review

Response: Thank you for your comments.

We have inserted tables in the manuscript.

2. Many language errors

Response: Thank you for your comments. 

We send the revised file back to the translation agency for revision in English (Attachment 4).

3. Many abbreviation don’t correspond correctly to their phrases Like “RCW” and “RHC” 

Response: Thank you for your comments.

We have revised the abbreviations of RCW and RHC in the text to make it the same as the full text. RCW, respiratory care ward; RHC, respiratory home care

4. “The main purpose of this study is to provide a solution for the shortage of ICU beds” That is a confusing sentence as this isn’t the aim of the study!! 

Response: Thank you for your comments. 

We have deleted this sentence from the introduction.

---

## [Decision Letter · Decision Letter 1]

20 Oct 2021

PONE-D-21-24193R1Stress on caregivers providing prolonged mechanical ventilation patient care in different facilities: A cross-sectional studyPLOS ONE

Dear Dr. Tsai,

Thank you for submitting your manuscript to PLOS ONE. After careful consideration, we feel that it has merit but does not fully meet PLOS ONE’s publication criteria as it currently stands. Therefore, we invite you to submit a revised version of the manuscript that addresses the points raised during the review process. Please revise.

We look forward to receiving your revised manuscript.

Kind regards,

Academic Editor

PLOS ONE

Reviewers' comments:

Reviewer's Responses to Questions

**Comments to the Author**

1. If the authors have adequately addressed your comments raised in a previous round of review and you feel that this manuscript is now acceptable for publication, you may indicate that here to bypass the “Comments to the Author” section, enter your conflict of interest statement in the “Confidential to Editor” section, and submit your "Accept" recommendation.

Reviewer #1: All comments have been addressed

Reviewer #2: (No Response)

2. Is the manuscript technically sound, and do the data support the conclusions?

Reviewer #1: Yes

Reviewer #2: No

3. Has the statistical analysis been performed appropriately and rigorously? 

Reviewer #1: Yes

Reviewer #2: No

4. Have the authors made all data underlying the findings in their manuscript fully available?

Reviewer #1: Yes

Reviewer #2: No

5. Is the manuscript presented in an intelligible fashion and written in standard English?

Reviewer #1: Yes

Reviewer #2: No

6. Review Comments to the Author

Reviewer #1: Thank you for taking time to address the comments. I feel, there are a few corrections needed.

Abstract

Page 2, line 42-45, the sentence lacks subject

Page 3, line 58, caregivers home or in-home caregivers?

Introduction

Line 70, Studies…… This sentence is not complete.

Reviewer #2: 1. “Home-based interviewees reported that taking care of

the patient of their family worsens the relationship between family members than the

intensive care unit group interviewees”

What do you mean exactly by this result in the abstract, are you pointing to family member giving care to chronic ventilated patients and the consequences or the respiratory caregiver and the relation to the family ?!

2. In the abstract after that sentence you switch to stress as consequence of this relation to family

You conclude in the abstract that “primary family caregivers of patients with

prolonged mechanical ventilation had the highest level of stress among in-home

caregivers”

I assume those are respiratory therapist handling patients at home as you r inclusion criteria is of healthcare professional only not family members taking care of relative on home ventilator, but in the results you mention nothing about their stress level, only you mention about their relation with family members

3. By the end your abstract isn’t well written and the results aren’t summarized in good way for the reader to perceive the message.

4.Your population isn’t clear or defined especially regarding those involved in RHC (respiratory home care)

5.The objective of your study isn’t clear

“We measured aspects related to their lives using a Likert scale”

6.Where’s the detailed table showing the numbers of respondents in the different subcategories based on the variables you measured , it seems based on table one that all or most of population were from RHC

7. PLOS authors have the option to publish the peer review history of their article (what does this mean?). If published, this will include your full peer review and any attached files.

Reviewer #1: **Yes: **Dr. Shital Adhikari, DM (Pulmonary, Critical Care and Sleep Medicine)

Reviewer #2: **Yes: **aljamaan, fadi

---

## [Author Response · Author response to Decision Letter 1]

29 Dec 2021

Response to reviewers’ comments

Reviewer #1:

1. Abstract: Page 2, line 42-45, the sentence lacks subject

Response: Thanks for your queries. 

We found no statistically significant difference at p=0.05, so we deleted this sentence in the abstract section.

2. Abstract: Page 3, line 58, caregivers home or in-home caregivers?

Response: Thanks for your comments. 

Since what we are visiting is to take care of the family members of PMV patients at home, this should be in-home caregiver. We revise the sentence as “Thus, the government should provide more support systems for in-home caregivers to relieve their life stress.” (Page 3, line 50-51 in revised vision)

3. Introduction: Line 70, Studies…… This sentence is not complete.

Response: Thanks for your comments. 

We are grateful for this suggestion and have completed sentences to clarify this point as follows: “Studies show that as patients who stay at home are more familiar with the environment, they feel more comfortable, and experience is better than the in-house care patients. [14].” (Page 5, line 76-78 in revised vision)

 

Reviewer #2: 

1. “Home-based interviewees reported that taking care of the patient of their family worsens the relationship between family members than the intensive care unit group interviewees” What do you mean exactly by this result in the abstract, are you pointing to family member giving care to chronic ventilated patients and the consequences or the respiratory caregiver and the relation to the family?! 

Response: Thanks for your queries. 

We have revised the result of abstract as follows: “As the results of the life impact presented in the family domain that RHC interviewees (OR=2.54) feel it worsens family relationships than the ICU group; however, there is less psychological stress in RCC (OR=0.54) and RCW (OR=0.16) than ICU group. In the social domain that RHC interviewees feel reducing friend/family interactivity (OR=2.18) and community/religious activities (OR=2.06) than ICU group; however, RCW interviewees fell less effect with leisure time (OR=0.37) and work time (OR=0.41) than ICU group. Furthermore, the economic domain of reducing family income because caregivers cannot work is less influenced in RCW interviewees (OR=0.43) than in ICU groups.” (Page 2, line 39-48 in revise vision)

2. In the abstract after that sentence you switch to stress as consequence of this relation to family You conclude in the abstract that “primary family caregivers of patients with prolonged mechanical ventilation had the highest level of stress among in-home caregivers ”I assume those are respiratory therapist handling patients at home as you r inclusion criteria is of healthcare professional only not family members taking care of relative on home ventilator, but in the results you mention nothing about their stress level, only you mention about their relation with family members 

Response: Thanks for your queries.

We have revised the conclusion in abstract as follows: “Stress levels of primary family caregivers of patients with prolonged mechanical ventilation had the highest stress level among in-home caregivers and less in the RCW group. Thus, the government should provide more support systems for in-home caregivers to relieve their life stress.” (Page 2-3, line 66-69 in revised vision)

3. By the end your abstract isn’t well written and the results aren’t summarized in good way for the reader to perceive the message. 

Response: Thanks for your comments.

We have major revision in the abstract (Page 2-3 line27-51, in revised vision).

4. Your population isn’t clear or defined especially regarding those involved in RHC (respiratory home care) 

Response: Thanks for your comments.

We have revised the introduction as follows: 

“The regulations of IPP are fee for service for ICU care (within 21 d), per-diem for RCC (for up to 42 d), per capitation for RCW, and monthly payment for home ventilation service. The ICU, RCC, and RCW belong to the institution care, and 24-hours medical staff is available. The patients who use ventilators at home have nurses’ and respiratory therapists’ visits twice a month and physicians’ visits once two months.” (Page 4 line 69-73, in revised vision)

5. The objective of your study isn’t clear “We measured aspects related to their lives using a Likert scale” 

Response: Thanks for your queries. 

We have revised the part of questionnaire development in Materials and Methods section as follows:

“The questionnaires have three domains (family, social, and economic aspects) that were measured related to their lives using a Likert scale (from "strongly agree" to "strongly disagree") to evaluate the impact of the IPP and the stress of life on main caregivers of PMV patients at different units.” (Page 6 line 102-105, in revised vision)

6. Where’s the detailed table showing the numbers of respondents in the different subcategories based on the variables you measured, it seems based on table one that all or most of population were from RHC. 

Response: Thanks for your queries. 

The numbers of respondents in the different subcategories were showed in table 2. A total of 601 interviewees were analyzed, and they consisted of ICU (150), RCC (150), RCW(150), and RHC(151). (Page 9-10, table 2, line 182-185, in revised vision)

 

Response to the Editor’s comments

1. Please include captions for your Supporting Information files at the end of your manuscript, and update any in-text citations to match accordingly. Please see our Supporting Information guidelines for more information: http://journals.plos.org/plosone/s/supporting-information.

Response: Thanks for your comments. 

We added our supporting information at the end of our manuscript, and please see line 290-296 on page 20. The supporting information is as follows:

S1 File: Question about the impact of taking care of prolonged mechanical ventilation patients on life (English and Chinese version)

S2 File: Interviewees Manual (English and Chinese version)

S3: 64 Interviewee's institutions information

S4: The impact on the family caregivers’ life between the ICU, RCC, RCW and RHC groups

2. We note your Data Availability statement:

"All relevant data are within the table 1-5, the raw data can’t share public due to the data contain personal information. This study was approved by the Institutional Review Board of China Medical University Hospital (IRB No. CMUH102-REC3-105), Taiwan." According to PLOS policy, when specific legal or ethical restrictions prohibit public sharing of a data set, authors must indicate how others may obtain access to the data. Authors must share the “minimal data set” for their submission. PLOS defines the minimal data set to consist of the data required to replicate all study findings reported in the article, as well as related metadata and methods. Please see this link for more details: https://journals.plos.org/plosone/s/data-availability#loc-minimal-data-set-definition If the minimal data set is contained within the raw data, please update your Data Availability statement with a non-author contact who can provide these data upon request. This could be a contact at the Institutional Review Board of China Medical University Hospital, Taiwan, or other such body 

Response: Thanks for your comments. 

We have added the supporting file (S4 file) for raw data of the figure 1-3. If you have any questions related to the minimal data set, please let us know.

Author’s Statement 

1. We are sorry that we found that the first question of the economic domain in figure 3 was misquoted the fifth question of the social domain in Table 3. We have revised it.

---

## [Decision Letter · Decision Letter 2]

31 Jan 2022

PONE-D-21-24193R2Stress on caregivers providing prolonged mechanical ventilation patient care in different facilities: A cross-sectional studyPLOS ONE

Dear Dr. Tsai,

Thank you for submitting your manuscript to PLOS ONE. After careful consideration, we feel that it has merit but does not fully meet PLOS ONE’s publication criteria as it currently stands. Therefore, we invite you to submit a revised version of the manuscript that addresses the points raised during the review process. Please address the issues and revise accordingly.

We look forward to receiving your revised manuscript.

Kind regards,

Academic Editor

PLOS ONE

Reviewers' comments:

Reviewer's Responses to Questions

**Comments to the Author**

1. If the authors have adequately addressed your comments raised in a previous round of review and you feel that this manuscript is now acceptable for publication, you may indicate that here to bypass the “Comments to the Author” section, enter your conflict of interest statement in the “Confidential to Editor” section, and submit your "Accept" recommendation.

Reviewer #1: All comments have been addressed

Reviewer #2: All comments have been addressed

2. Is the manuscript technically sound, and do the data support the conclusions?

Reviewer #1: Yes

Reviewer #2: No

3. Has the statistical analysis been performed appropriately and rigorously? 

Reviewer #1: Yes

Reviewer #2: No

4. Have the authors made all data underlying the findings in their manuscript fully available?

Reviewer #1: Yes

Reviewer #2: No

5. Is the manuscript presented in an intelligible fashion and written in standard English?

Reviewer #1: Yes

Reviewer #2: No

6. Review Comments to the Author

Reviewer #1: Dear Authors

Thank you for addressing the comments. The evidence you have generated about the level of stress of caregivers of patients on prolonged mechanical ventilation (PMV) in different settings will be very useful for planning of care.

Reviewer #2: The paper is still confusing , and mixed up , still you confuse the family caregiver with the respiratory therapist in your abstract and manuscript, to the degree I am not sure the result belong to whom , please revisit your conclusion in the abstract, this is after 2 runs of revision

I cant get the theme from the paper and conclude main outcome or idea

7. PLOS authors have the option to publish the peer review history of their article (what does this mean?). If published, this will include your full peer review and any attached files.

Reviewer #1: **Yes: **Dr. Shital Adhikari

Reviewer #2: **Yes: **aljamaan, fadi

---

## [Author Response · Author response to Decision Letter 2]

15 Mar 2022

Response to Reviewer Comments:

1. Reviewer #2: The paper is still confusing, and mixed up, still you confuse the family caregiver with the respiratory therapist in your abstract and manuscript, to the degree I am not sure the result belong to whom, please revisit your conclusion in the abstract, this is after 2 runs of revision. I cant get the theme from the paper and conclude main outcome or idea.

Ans.: We feel very sorry for the confusion. We have revised and marked with red color for revision in the manuscript. Thank you for reminding and providing the opportunity to revise.

Abstract (Revised) 

Purpose: Taiwan has implemented an integrated prospective payment program (IPP) for prolonged mechanical ventilation (PMV) patients that consists of four stages of care: intensive care unit (ICU), respiratory care center (RCC), respiratory care ward (RCW), and respiratory home care (RHC). We aimed to investigate the life impact on family caregivers of PMV patients opting for a payment program and compared different care units. Method: A total of 610 questionnaires were recalled. Statistical analyses were conducted by using the chi-square test and multivariate logistic regression model. Results: The results indicated no associations between caregivers’ stress levels and opting for a payment program. Participants in the non-IPP group spent less time with friends and family owing to caregiver responsibilities. The results of the family domain show that the RHC group (OR=2.54) had worsened family relationships compared with the ICU group; however, there was less psychological stress in the RCC (OR=0.54) and RCW (OR=0.16) groups than in the ICU group. In the social domain, RHC interviewees experienced reduced friend and family interactivity (OR=2.18) and community or religious activities (OR=2.06) than the ICU group. The RCW group felt that leisure and work time had less effect (OR=0.37 and 0.41) than the ICU group. Furthermore, RCW interviewees (OR=0.43) were less influenced by the reduced family income than the ICU group in the economic domain. Conclusions: RHC family caregivers had the highest level of stress, whereas family caregivers in the RCW group had the lowest level of stress.

---

## [Decision Letter · Decision Letter 3]

4 Apr 2022

PONE-D-21-24193R3Stress on caregivers providing prolonged mechanical ventilation patient care in different facilities: A cross-sectional studyPLOS ONE

Dear Dr. Tsai,

Thank you for submitting your manuscript to PLOS ONE. After careful consideration, we feel that it has merit but does not fully meet PLOS ONE’s publication criteria as it currently stands. Therefore, we invite you to submit a revised version of the manuscript that addresses the points raised during the review process.

Please revise.

We look forward to receiving your revised manuscript.

Kind regards,

Academic Editor

PLOS ONE

Journal Requirements:

Reviewers' comments:

Reviewer's Responses to Questions

**Comments to the Author**

1. If the authors have adequately addressed your comments raised in a previous round of review and you feel that this manuscript is now acceptable for publication, you may indicate that here to bypass the “Comments to the Author” section, enter your conflict of interest statement in the “Confidential to Editor” section, and submit your "Accept" recommendation.

Reviewer #1: All comments have been addressed

Reviewer #3: All comments have been addressed

2. Is the manuscript technically sound, and do the data support the conclusions?

Reviewer #1: Yes

Reviewer #3: Yes

3. Has the statistical analysis been performed appropriately and rigorously? 

Reviewer #1: Yes

Reviewer #3: Yes

4. Have the authors made all data underlying the findings in their manuscript fully available?

Reviewer #1: Yes

Reviewer #3: Yes

5. Is the manuscript presented in an intelligible fashion and written in standard English?

Reviewer #1: Yes

Reviewer #3: Yes

6. Review Comments to the Author

Reviewer #1: The article will be very useful for deciding site of care for patients who need long term ventilatory support. The findings may serve as issues for discussion.

Reviewer #3: Line 66

“The ICU, RCC, and RCW is belonging the institution” –- suggest English consultation

Line 97

“The questionnaires have three domains (…) was measured aspects related to their lives by a Likert scale(…) to compared with the Impact of the IPP and different unit of main caregiver of PMV patients on the stress of life. ” –- suggest English consultation

Line 113

“the acute ICU stage was the time or less than 7 days in RCC after transfer from ICU;”

→- “the acute ICU stage was the time in ICU and/or less than 7 days in RCC after transfer from ICU;”---???

Line 143

“The monthly expense for the IPP group was significantly lower than that for the non-IPP group (USD 745 vs. USD 912 respectively) (Table 1).” --- The definition of “monthly expense” is not clear. Is it the expense paid by the family or by the Taiwan National Health Insurance Program?

Line 144

“the non-IPP group” --- The recruitment of “non-IPP group” was not described in the methodology.

Line 148 Table 1 --- The last line of 1st column

“Average monthly expense

(SD) (USD) “

→“Average monthly expense (USD)

Mean (SD) “

Line 193, 208, 216

“Odds of agree in the life impact” →“Odds of agreement in the life impact”

7. PLOS authors have the option to publish the peer review history of their article (what does this mean?). If published, this will include your full peer review and any attached files.

Reviewer #1: No

Reviewer #3: No

---

## [Author Response · Author response to Decision Letter 3]

23 Apr 2022

Response to reviewers’ comments:

PONE-D-21-24193R3

Stress on caregivers providing prolonged mechanical ventilation patient care in different facilities: A cross-sectional study

Reviewer #1: 

The article will be very useful for deciding site of care for patients who need long term ventilatory support. The findings may serve as issues for discussion.

Ans.: Thanks for your comments.

Reviewer #3:

1. Line 66:“The ICU, RCC, and RCW is belonging the institution” –- suggest English consultation “

Ans.: Thanks for your comments.

We are grateful for this suggestion and try to clarify this sentence, which was as follows “The ICU, RCC, and RCW belong to the institutions,” (Page 3, line 67 in revised vision)

2. Line 97:“The questionnaires have three domains (…) was measured aspects related to their lives by a Likert scale(…) to compared with the Impact of the IPP and different unit of main caregiver of PMV patients on the stress of life. ” –- suggest English consultation

Ans.: Thanks for your comments.

We are grateful for this suggestion and try to clarify this sentence, which was as follows “The questionnaires have three domains (family, social, and economic). They measured aspects related to their lives by a Likert scale (from strongly agree to strongly disagree) to compare the impact of the IPP and different units of primary caregivers of PMV patients on the stress of life.” (Page 5, line 97-100 in revised vision)

3. Line 113:“the acute ICU stage was the time or less than 7 days in RCC after transfer from ICU;”→-“the acute ICU stage was the time in ICU and/or less than 7 days in RCC after transfer from ICU;”---???

Ans.: Thanks for your comments. 

We are grateful for this suggestion and try to clarify this sentence, which was as follows “the acute ICU stage was the time in ICU and/or less than 7 days in RCC after transfer from ICU” (Page 5, line 113-114 in revised vision).

4. Line 143:“The monthly expense for the IPP group was significantly lower than that for the non-IPP group (USD 745 vs. USD 912 respectively) (Table 1).” --- The definition of “monthly expense” is not clear. Is it the expense paid by the family or by the Taiwan National Health Insurance Program? 

 Ans.: Thanks for your comments. 

We are grateful for this suggestion and try to clarify this sentence, which was as follows “The monthly expense paid by the family for the IPP group was significantly lower than that for the non-IPP group (USD 745 vs. USD 912, respectively) (Table 1).” (Page 7, line 145 in revised vision).

5. Line 144:“the non-IPP group” --- The recruitment of “non-IPP group” was not described in the methodology.

Ans.: Thanks for your comments.

We are grateful for this suggestion and have added sentences to clarify this

point, which were as follows “The participants were divided into two groups (IPP and none-IPP group) according to whether they had joined IPP.” (Page 5, line 117-118 in revised vision).

6. Line 148 Table 1 --- The last line of 1st column“Average monthly expense(SD) (USD) “→“Average monthly expense (USD) Mean (SD)

Ans.: Thanks for your comments. We have changed the description. (Page 8, line 151 the last line of 1st column in revised vision).

7. Line 193, 208, 216 “Odds of agree in the life impact” →“Odds of agreement in the life impact”

Ans.: Thanks for your suggestion. We have changed the description. (Page 15, line 196; Page 16, line 211; Page 16, line 219 in revised vision).

---

## [Decision Letter · Decision Letter 4]

11 May 2022

Stress on caregivers providing prolonged mechanical ventilation patient care in different facilities: A cross-sectional study

PONE-D-21-24193R4

Dear Dr. Tsai,

We’re pleased to inform you that your manuscript has been judged scientifically suitable for publication and will be formally accepted for publication once it meets all outstanding technical requirements.

Kind regards,

Academic Editor

PLOS ONE

Additional Editor Comments (optional):

Reviewers' comments:

Reviewer's Responses to Questions

**Comments to the Author**

1. If the authors have adequately addressed your comments raised in a previous round of review and you feel that this manuscript is now acceptable for publication, you may indicate that here to bypass the “Comments to the Author” section, enter your conflict of interest statement in the “Confidential to Editor” section, and submit your "Accept" recommendation.

Reviewer #1: All comments have been addressed

Reviewer #3: All comments have been addressed

2. Is the manuscript technically sound, and do the data support the conclusions?

Reviewer #1: Yes

Reviewer #3: Yes

3. Has the statistical analysis been performed appropriately and rigorously? 

Reviewer #1: Yes

Reviewer #3: Yes

4. Have the authors made all data underlying the findings in their manuscript fully available?

Reviewer #1: Yes

Reviewer #3: Yes

5. Is the manuscript presented in an intelligible fashion and written in standard English?

Reviewer #1: Yes

Reviewer #3: Yes

6. Review Comments to the Author

Reviewer #1: The article has been interesting showing the stress level when one has to take care of close relatives when being managed with ventilatory support in different conditions. Thank you for addressing all the concerns raised

Reviewer #3: Line 50

“The PMV incidence rates in Europe and the US were found to be 5 %-11 % [3-6]. Taiwan had a PMV rate of 20 %, while China had a PMV rate of 36.1 %,”

---- The background information of incidence rate calculation is not provided. Are these incidence rates based on per 100 ventilated ICU admissions?

Line 138

“As comprehensive analyses datasets were generated by a previous study [26], the

current dataset was generated from 687 eligible respondents from 64 institutions (6

medical centers, 11 regional hospitals, 23 district hospitals, 15 nursing homes, and 9

home care centers) were included in this study (S3 File).”

---- English consultation.

Line 183----Table 5

“Disagree”→ “disagree”

7. PLOS authors have the option to publish the peer review history of their article (what does this mean?). If published, this will include your full peer review and any attached files.

Reviewer #1: No

Reviewer #3: No

---

## [Editor Report · Acceptance letter]

16 May 2022

PONE-D-21-24193R4 

Stress on caregivers providing prolonged mechanical ventilation patient care in different facilities: A cross-sectional study 

Dear Dr. Tsai:

I'm pleased to inform you that your manuscript has been deemed suitable for publication in PLOS ONE. Congratulations! Your manuscript is now with our production department. 

Kind regards, 

on behalf of

Dr. Robert Jeenchen Chen 

Academic Editor

PLOS ONE